# GRACE Combined with WSD to Assess the Change in Drought Severity in Arid Asia

Jiawei Liu [1,2,3,4], Guofeng Zhu [1,2,3,4,*], Kailiang Zhao [1,2,3,4], Yinying Jiao [1,2,3], Yuwei Liu [1,2,3], Mingyue Yang [1], Wenhao Zhang [1,2,3], Dongdong Qiu [1,2,3], Xinrui Lin [1,2,3] and Linlin Ye [1,2,3]

[1] College of Geography and Environmental Science, Northwest Normal University, Lanzhou 730070, China; 2021222908@nwnu.edu.cn (J.L.); 2020222791@nwnu.edu.cn (K.Z.); 201875110110@nwnu.edu.cn (Y.J.); 2020212686@nwnu.edu.cn (Y.L.); 2021222973@nwnu.edu.cn (M.Y.); 2021222902@nwnu.edu.cn (W.Z.); 2021212816@nwnu.edu.cn (D.Q.); 2021212820@nwnu.edu.cn (X.L.); 2021222917@nwnu.edu.cn (L.Y.)
[2] Key Laboratory of Resource Environment and Sustainable Development of Oasis, Lanzhou 730070, China
[3] Shiyang River Ecological Environment Observation Station, Northwest Normal University, Lanzhou 730070, China
[4] Lanzhou Sub-Center, Remote Sensing Application Center, Ministry of Agriculture, Lanzhou 730000, China
* Correspondence: zhugf@nwnu.edu.cn

**Abstract:** Gravity Recovery and Climate Experiment (GRACE) satellite data are widely used in drought studies. In this study, we quantified drought severity based on land terrestrial water storage (TWS) changes in GRACE data. We used the water storage deficit (WSD) and water storage deficit index (WSDI) to identify the drought events and evaluate the drought severity. The WSDI calculated by GRACE provides an effective assessment method when assessing the extent of drought over large areas under a lack of site data. The results show a total of 22 drought events in the central Asian dry zone during the study period. During spring and autumn, the droughts among these incidents occurred more frequently and severely. The longest and most severe drought occurred near the Caspian Sea. In the arid area of central Asia, the north of the region tended to be moist (the WSDI value was 0.04 year$^{-1}$), and the south, east, and Caspian Sea area tended to be drier (the WSDI values were $-0.07$ year$^{-1}$ in the south, $-0.11$ year$^{-1}$ in the east, and $-0.19$ year$^{-1}$ in the Caspian Sea). These study results can provide a key scientific basis for agricultural development, food security, and climate change response in the Asian arid zone.

**Keywords:** central Asia; drought; degree of drought; GRACE; WSDI





## 1. Introduction

Drought is one of the most important natural disasters in the world; it not only affects human activities but also has a negative impact on the environment, agriculture, and economic development. Monitoring and quantifying drought-induced changes in water storage are of positive significance for assessing the extent of drought. Moreover, under the current rapid economic development and population increase scenario, a more scientific assessment of the change in drought extent is necessary [1] (Long et al., 2013). Water shortages will be even worse in a decade's time, affecting as much as half the world's population [2] (Dharpure et al., 2020). Currently, climate change is becoming more unstable, and the negative effects of drought are likely to intensify [3] (Gerdener et al., 2020). The Asian arid zone is located in the hinterland of the continent, with less drought and more rain, including the widespread Gobi Desert. The ecological environment here is very fragile, and climate change is very significant. The Asian arid zone has been under serious pressure from climate change and the frequent occurrence of extreme climate events [4] (Trenberth et al., 2014). Therefore, it is very important to accurately assess the degree of aridity in arid and semi-arid regions.

However, the quantification of drought and its hydrological effects remains a major challenge due to the limitations of monitoring means. The Gravity Recovery and Climate Experiment (GRACE) launch generated a new tool for monitoring and assessing drought events. After GRACE observations, it was concluded that the gravity change mainly resulted from the large-scale movement of surface water, and the GRACE signal was transformed into variation in the Total Water Storage Anomaly (TWSA) on land [5] (Hu et al., 2019). The terrestrial water storage (TWS) derived from GRACE is an important indicator for quantifying changes in surface water and groundwater storage on land [6] (Wahr and John, 2004) and is widely used to monitor hydrological changes caused by drought. The TWSA retrieved by GRACE includes all forms of water stored above and below the surface, including snow, surface water, soil moisture, and groundwater [7] (Wu et al., 2021), and can thus serve as an effective substitute for hydrological information.

Based on the GRACE TWSA, several drought indices have been put into use and developed to date to elaborate drought assessment. Yirdaw et al. [8] (Yirdaw et al., 2008) proposed the total storage deficit index (TSDI) by using the TWSA value of GRACE and the Palmer drought Severity Index (PDSI) and soil water deficit index (SMDI). Thomas et al. [9] (Thomas et al., 2014) proposed a quantitative method to measure the occurrence and severity of hydrological drought based on GRACE data. Sinha et al. [10] (Sinha et al., 2017) further extended the method of measuring the occurrence of hydrological drought based on GRACE data and designed a drought index, the water storage deficit index (WSDI), using the TWS changes estimated by GRACE to quantify the intensity and severity of drought. Hosseini-moghari et al. [11] (Hosseini-moghari et al., 2019) developed an improved total storage deficit index (MTSDI) using the remaining time series of TWSA to eliminate the influence of human factors on TWSA changes. Unlike the traditional drought index, the WSDI is calculated from the change in terrestrial water storage, which can better show the dynamics of drought formation. In addition, when there is a need to assess the extent of drought over a large area and there is a lack of data from meteorological or hydrological stations in some areas, the WSDI is based on the integrated land water storage changes observed by satellites, so the storage amount can be monitored at any time without the limitations of traditional observations.

In recent years, drought has become more severe and has frequently occurred in many parts. In this study, we evaluated the degree of drought in Asia and Europe based on GRACE data. Our aim was to (1) clarify the overall change trend of the drought degree in different regions of arid Asia and to (2) analyze the influencing factors leading to the change in drought degree. As a classical case, this study provides an evaluation of the drought degree in an arid region, which can form a scientific basis to facilitate adjusting agricultural policy and coping with climate change in that area.

## 2. Material and Methods

### 2.1. Research Data

The space research center at the University of Texas (CSR), the German Research Centre for Geosciences (GFZ), and the Jet Propulsion Laboratory (JPL) provide three GRACE/GRACE-FO spherical harmonic (SH) solution land storage (RL06) datasets that were used in this study [12] (J. Kusche et al., 2009). The water storage data obtained from these datasets are expressed in centimeter Equivalent Water Height (EWH) [13] (Landerer and Swenson, 2012). The three organizations performed several post-processing corrections to their SH solutions to isolate surface impound signals. The $C_{20}$ spherical harmonic coefficient was replaced by satellite laser-ranging solutions [14] (Cheng, 2011). A method by Swenson et al. [15] (Swenson et al., 2008) was used to estimate the first-order coefficients. A 300 km wide Gaussian filter was used to smooth the spherical harmonic solution to minimize correlation and high-frequency error. To reduce the error and make the data more accurate, the monthly TWS data provided by CSR, GFZ, and JPL were processed, extracted, and added to obtain the average value. All the missing data were filled with the average values of the months before and after the missing month using

the linear interpolation method. GRACE returned to Earth in July 2017, and the second-generation GRACE Follow (GRACE-FO) started operations in May 2018. To assess the drought situation more accurately in the arid regions of central Asia, in this paper, we used data taken from 16 years from January 2003 to December 2020, excluding 2017 and 2018. We defer to the meteorological division law, namely, that the spring months are March, April, and May; the summer months are June, July, and August; the autumn months are September, October, and November; and the winter months are December, January, and February.

### 2.2. GRACE Inversion

The principle of GRACE satellite data inversion is to calculate the change in the Earth's mass density at each moment by calculating the change in the spherical harmonic potential coefficient, which describes the time-varying gravity field and expresses the change in the mass density in the form of the equivalent water height. The Earth's gravity field is often expressed in terms of the spherical harmonics (SH) expression of the geoid that is the equipotential surface corresponding to the mean sea level. In general, the spherical harmonic expression of the geoid serves as the Earth's gravity field by the equipotential surface. Most remaining signals are relevant to changes in the TWS after removing atmospheric and ocean mass effects from the climate and ocean general circulation models. Therefore, terrestrial water storage anomalies (TWSA) $\Delta\eta_{land}$ on land can be directly described by gravity coefficient anomalies ($\Delta C_{lm}$, $\Delta S_{lm}$):

$$\Delta\eta_{land}(\theta, \lambda) = \frac{a\rho_{ave}}{3\rho_w} \sum_{l=0}^{\infty} \sum_{m=0}^{l} \widetilde{P}_{lm}(\cos\theta) \frac{2l+1}{1+k_l} (\Delta C_{lm}\cos(m\lambda) + \Delta S_{lm}\sin(m\lambda)) \quad (1)$$

where $\Delta\eta_{land}$ is the Equivalent Water Height (EWH); $a$ is the average radius of the Earth; $\Delta C_{lm}$ and $\Delta S_{lm}$ are the variation values of spherical harmonic potential coefficients of the Earth's gravitational field; $\rho_w$ is water density; $\rho_{ave}$ represents the average density of the Earth; $l$ and $m$ are, respectively, the order and DEGREE of the spherical harmonic function; $C_{lm}$, $S_{lm}$ are the regularized spherical harmonic potential coefficients in the Earth's gravitational field; $\theta$ is the polar distance; $\lambda$ is longitude; $\widetilde{P}_{lm}$ is a fully normalized Legendre function; and $k_l$ is the first-order load Love number.

### 2.3. Estimation of Insufficient Water Storage Based on GRACE

The water storage deficit (*WSD*) can be calculated by obtaining the *TWSA* time series from GRACE [9] (Thomas et al., 2014), as shown below:

$$WSD_{i,j} = TWSA_{i,j} - \overline{TWSA_j} \quad (2)$$

In the formula, $TWSA_{i,j}$ denotes month $j$ of year i of the *TWSA* time series obtained by GRACE, and $\overline{TWSA_j}$ is the long-term (January 2003 to December 2020) mean of *TWSA* for the same month (the average of 16 values in month $j$). A negative *WSD* means that, compared with the monthly average, insufficient land water storage has led to a water storage deficit, while a positive value means that there is surplus water storage. A drought event is defined as a negative *WSD* lasting three months. To better assess *WSD*-based drought, this parameter translates to the water storage deficit index (WSDI) as follows:

$$WSDI = \frac{WSD - \mu}{\sigma} \quad (3)$$

where $\mu$ is the mean value of the *WSD* time series, and $\sigma$ represents the standard deviation of the *WSD* time series. The magnitude of the *WSDI* represents the drought intensity, and the classification of the drought intensity is shown in Table 1.

**Table 1.** Characterization of drought intensity by WSDI.

| WSDI Value | Drought Category |
|------------|------------------|
| W > 0 | No drought |
| $0 \geq W > -1$ | Mild drought |
| $-1 \geq W > -2$ | Moderate drought |
| $-2 \geq W > -3$ | Severe drought |
| $-3 \geq W$ | Extreme drought |

## 3. Study Area

The arid region of central Asia (34°34′–55°43′N, 46°48′–106°98′E) is located north of Pamir/the Qinghai–Tibet Plateau, south of the Urals/Altai Mountains, and east of the Caspian Sea and Volga River, and extends to Helan Mountain/Wushaoling mountains (Figure 1). It is the widest arid region in the temperate zone of the Earth's northern hemisphere [16] (Chen Xi et al., 2013). The arid areas are located in the hinterland of Eurasia, with a low proportion of maritime airflow reaching them. The upper westerlies transport weak water vapor from the remote Arctic and Atlantic Oceans. Due to the interception and uplift of vertical terrain, precipitation is concentrated in the mountains, and snow glaciers form in the mountains. The average annual precipitation in arid areas is less than 150 mm, and the spatial distribution is extremely unbalanced. The annual precipitation in mountainous areas is more than that in basins and plains [17] (Balashova Y. et al., 2006). The annual precipitation in forest areas of the Tianshan Mountains and the Altai Mountains reaches 1000 mm [18] (Hu Ruji et al., 2004). The annual precipitation in Turpan and Hami Basin is less than 100 mm [17] (Balashova Y. et al., 2006).

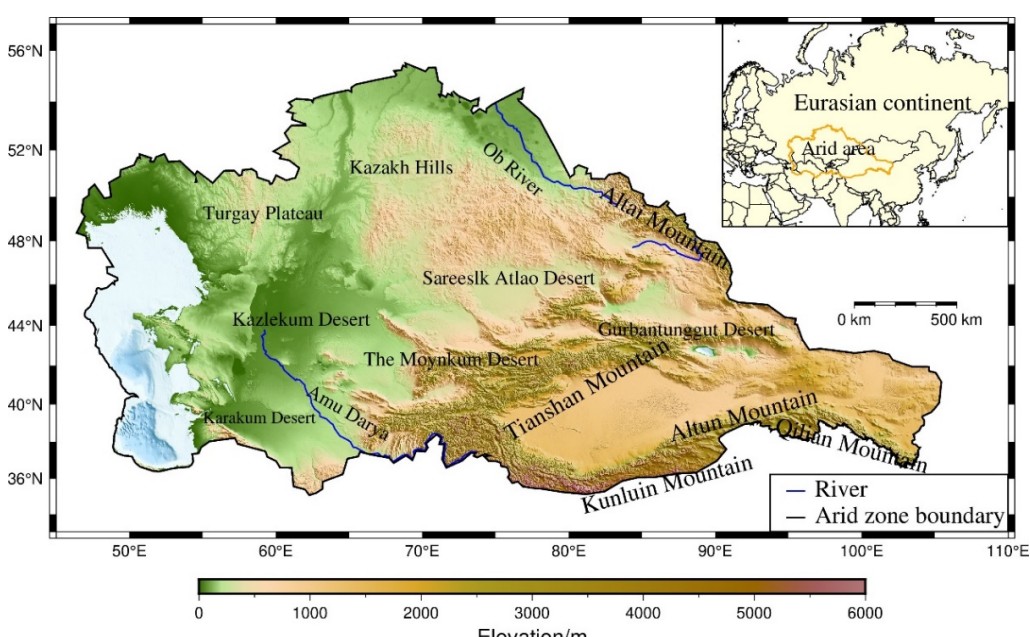

**Figure 1.** Overview map of the study area in the arid region of central Asia.

## 4. Results

### 4.1. Interannual Variation in TWS in the Arid Area

The TWS value in the vicinity of the Caspian Sea showed an overall downward trend from 2003 to 2020 but increased slightly from 2003 to 2005 and from 2015 to 2016 (Figures 2 and 3). The maximum value of TWS in the vicinity of the Caspian Sea was 141.10 mm, and the lowest value was −158.32 mm from 2003 to 2020. The TWS value near the Kazak Hills in the northern arid area fluctuated minutely from 2003 to 2012 and reached its lowest value of −51.77 mm in 2012. However, it rose continuously in the following five years, reaching

a peak value of 59.97 mm in 2019, and then decreased slightly to 37.78 mm in 2020. The TWS value of the southern Moynkum Desert in the arid area showed a small change range from 2003 to 2020 and increased slightly from 2003 to 2005, from 2008 to 2010, and from 2014 to 2019. After reaching a maximum value of 36.45 mm in 2005, the TWS value showed a downward trend. In 2008, the TWS value reached its lowest value, only −41.05 mm. The TWS value of the Gurbantünggüt Desert in the arid eastern area showed a downward trend and briefly increased from 2009 to 2011 and from 2014 to 2016. After that, the TWS value continued to decline and reached its lowest value of −39.52 mm in 2020.

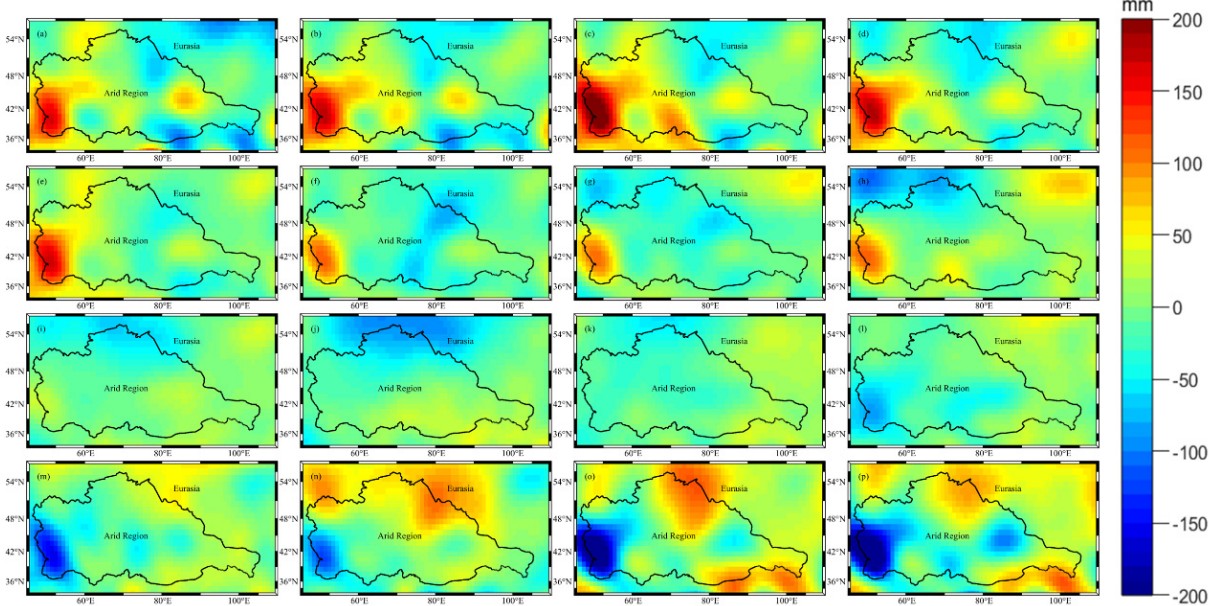

**Figure 2.** Annual variation of TWS in arid regions of Central Asia from 2003 to 2020.

### 4.2. Annual TWS Seasonal Variation in the Arid Area

According to GRACE gravity satellite data from 2003 to 2020, the spring, summer, autumn, and winter line maps of arid regions in central Asia were obtained (Figure 3). The TWS values of the Caspian Sea in spring, summer, autumn, and winter from 2003 to 2020 show a decreasing trend as a whole, with those in spring and summer being higher than those in autumn and winter. The average TWS values of the four seasons from 2003 to 2020 showed that autumn was always at the lowest value, only −37.30 mm, while the average TWS value in spring was the highest, up to 46.18 mm. In the vicinity of Kazak Hill, the TWS values in spring and winter from 2003 to 2020 were higher than those in summer and autumn. Among the average TWS values of the four seasons from 2003 to 2020, the TWS value in autumn was always the lowest, with an average value of −48.95 mm, while the average TWS value in spring was the highest, at 21.42 mm. In the southern part of the arid area, the TWS values in spring and summer were higher than those in autumn and winter. The TWS value in autumn was always the lowest, with an average value of −50.42 mm, while the TWS value in spring was the highest, at 31.66 mm. In the vicinity of the Gurbantünggüt Desert, the TWS values in the four seasons show a decreasing trend as a whole, and the TWS values in spring and summer were higher than those in autumn and winter, but small distinctions were observed in the average TWS values for the four seasons. The average TWS values in autumn were mostly at the lowest value, at −10.27 mm, and the highest value in spring was 7.93 mm.

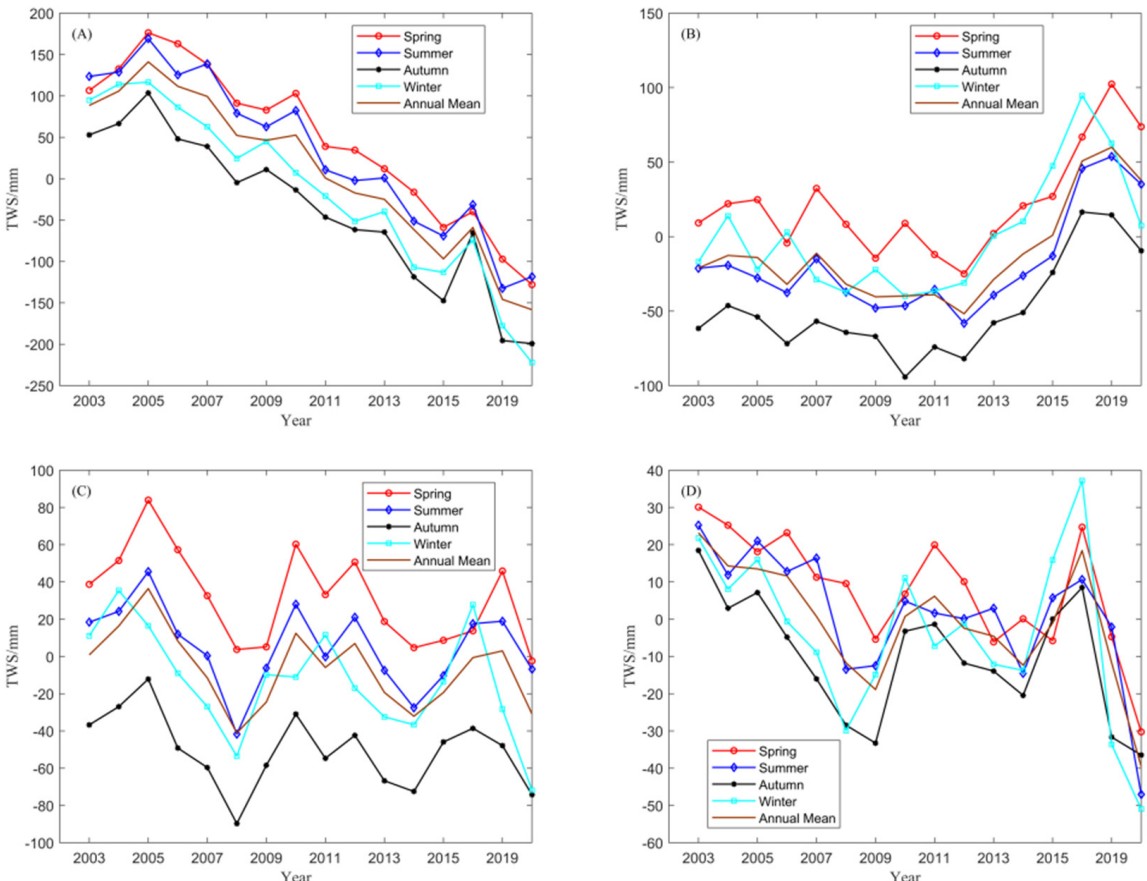

**Figure 3.** Seasonal variations in the (**A**) Caspian Sea area, (**B**) Kazakhskiy Melkosopochnik, (**C**) Moynkum Desert, and (**D**) Gurbantünggüt Desert TWS values in the arid region of central Asia from 2003 to 2020.

*4.3. Analysis of Drought Severity*

Land water storage deficits can visually represent the severity of drought. Figure 4 describes the water storage deficit (WSD) and drought events near the Caspian Sea, Kazakh Hills, Moynkum Desert, and Gurbantünggüt Desert in arid regions of central Asia from 2003 to 2020. A negative WSD value lasting three months or more is considered a drought event. These results show that significant water storage loss began to occur in the vicinity of the Caspian Sea in February 2011, which lasted until December 2020 and reached a maximum loss of −186.43 mm in December 2020, with a WSDI of −2.03. According to Table 1, a severe drought was occurring in the vicinity of the Caspian Sea. This indicates that Region A has been under drought since February 2011, and the degree of drought continues to increase. In the northern part of the arid area, there was a drought period of 78 months (from August 2007 to January 2014), and the maximum loss of water storage reached −53.78 mm (September 2010). In addition, there were four mild droughts or brief droughts (low degree of drought) in the Kazak Hills. There were three significant drought periods from October 2007 to December 2009, from February 2013 to August 2015, and from October 2019 to December 2020 in the southern Moynkum Desert, with the average deficit values reaching −25.37 mm, −20.42 mm, and −20.28 mm, respectively. The three droughts lasted 27, 31, and 15 months, respectively. There were three significant drought events in the Gurbantünggüt Desert in the arid eastern area from January 2003 to December 2020, among which the drought was the most serious from July 2019 to December 2020, with the average deficit reaching −30.78, the highest water storage deficit reaching −53.30, and a WSDI value of −2.56. According to Table 1, a severe drought was underway in the arid east. Figure 5 shows the WSDI time series and fitting trends of the four regions in the arid area of central Asia. The WSDI in the northern part of the arid area of central Asia shows

an overall upward trend (0.04 year$^{-1}$), while those for the Caspian Sea (−0.19 year$^{-1}$), south (−0.07 year$^{-1}$), and eastern (−0.11 year$^{-1}$) areas show a decreasing tendency. Trend analysis shows that water stress is worsening near the Caspian Sea, around the Moynkum Desert, and in the Gurbantünggüt Desert region, while water scarcity is improving near the Kazakh Hills.

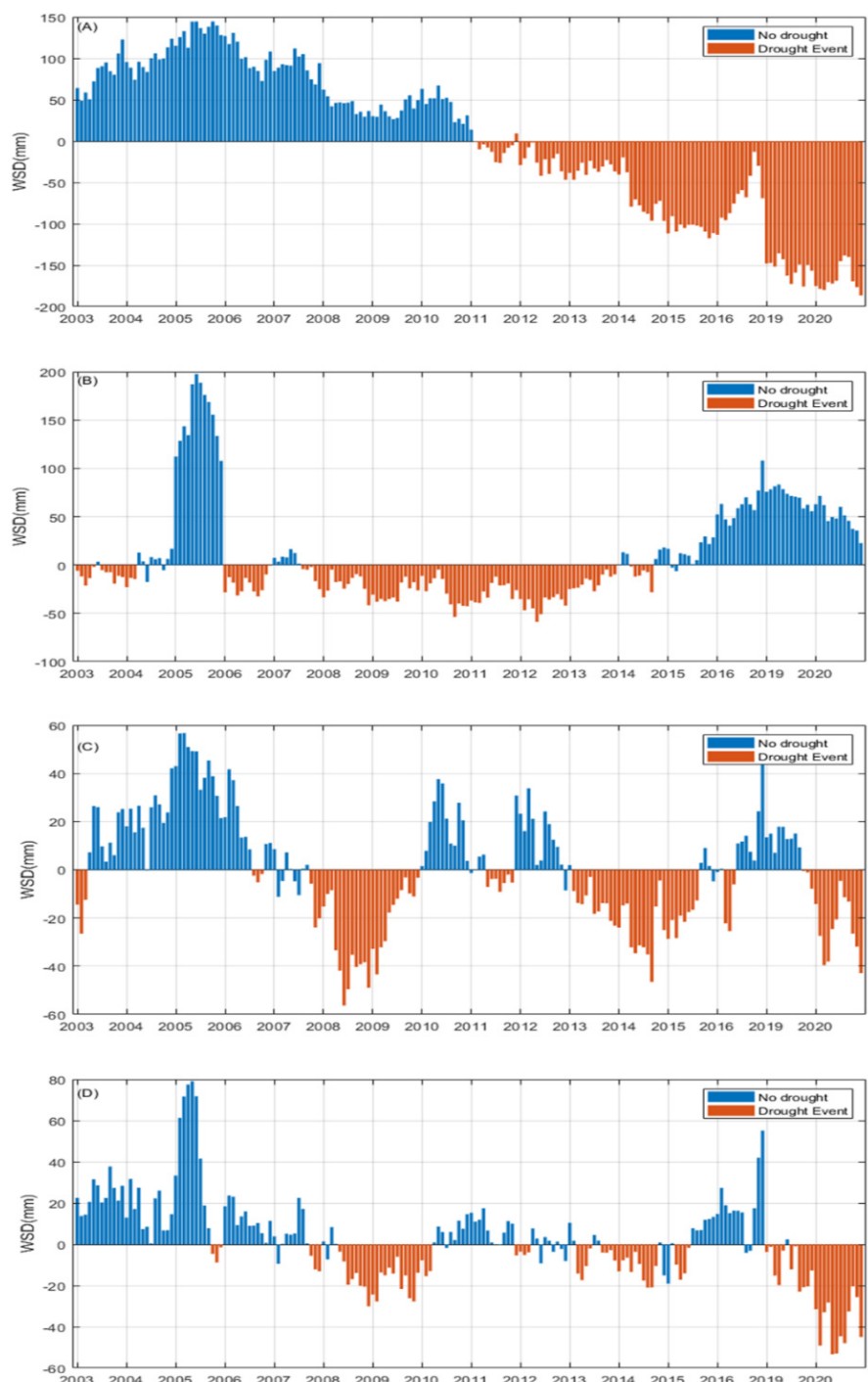

**Figure 4.** Statistics of the WSD and drought events near the (**A**) Caspian Sea, (**B**) Kazakh Hills, (**C**) Moynkum Desert, and (**D**) Gurbantünggüt Desert in arid regions of central Asia from 2003 to 2020.

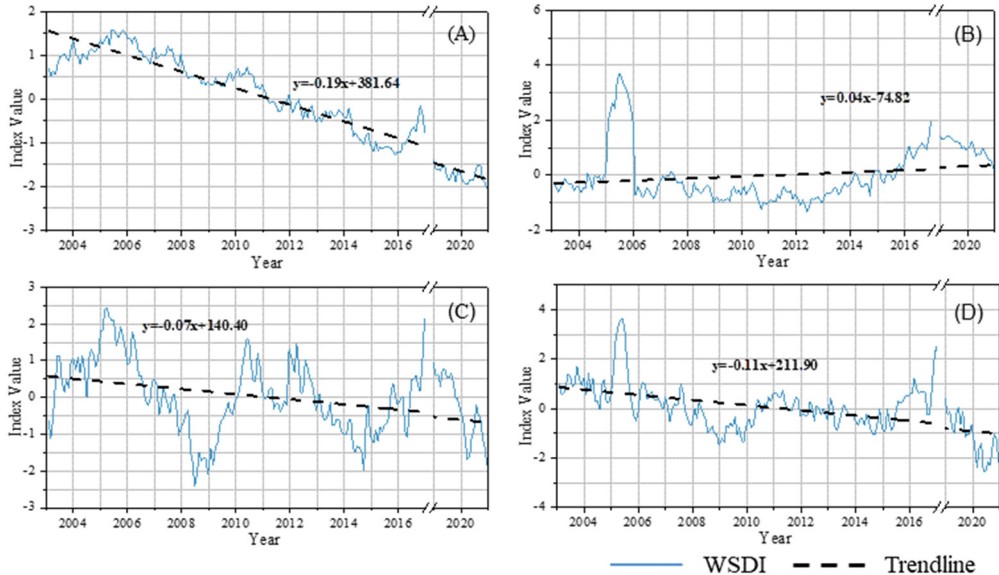

**Figure 5.** WSDI time series and fitting trends in the arid regions of central Asia near the (**A**) Caspian Sea, (**B**) Kazakh Hills, (**C**) Moynkum Desert, and (**D**) Gurbantünggüt Desert.

## 5. Discussion

### 5.1. Possible Impacts of Climate Change on Arid Asia

The frequency of drought incidents in Asia is subject to the fragile ecological environment, unstable precipitation, and evapotranspiration, which lead to land degradation, soil erosion, severe water shortage, and significant changes in local agricultural climate resources, directly threatening the livelihoods of hundreds of millions of local farmers and herdsmen [19] (Troy S., 2018). The drought problem widely concerns the meteorological community and other disciplines, especially in the arid region, where it is in reaction to global change. Previous studies have shown that the Asian drylands are sensitive to climate change to a much greater extent than other regions [20] (Xu et al., 2019). Dai et al. [21] (Dai et al., 2013) pointed out that in the context of global warming, the increase in water evapotranspiration in arid regions around the world may increase the severity of drying early in arid regions in Asia. The increase in evapotranspiration is caused by continuous high-temperature weather, which is exactly the main reason for the frequent occurrence of drought events in arid areas in recent years [22] (Lehner F et al., 2017). This area receives a small precipitation amount and has a large precipitation change rate. In addition, uneven annual distribution, dry air, and substantial surface evaporation are normally present. Climate change leads to irregular spatial and temporal distributions of water resources in arid areas of Asia, which aggravates the contradiction between water supply and demand. Grassland and herbage are susceptible to agricultural production and the scope of crop growth on account of their changing and expanding; meanwhile, the original balance of grassland degradation is broken too. Desertification areas are expanding, and extreme weather events are increasing, which has an important impact on the economy of arid regions in Asia. Therefore, in the face of the new challenges brought by climate change to the arid region of Asia, it is necessary to actively explore and fully understand climate change and its impact on the arid region of Asia, seek advantages and avoid disadvantages, develop an economic model suitable for the new form of the arid region of Asia, and reduce the huge losses brought by climate change to the arid region of Asia.

### 5.2. Uncertainty in Drought Severity Assessment

Climate and terrestrial hydrological conditions determine the severity of drought. Drought indices are used to measure the degree of drought, reflecting its cause, extent, beginning, end, and duration. Since drought affects a wide range of areas, agricultural, hydrological, and meteorological scientists have conducted in-depth studies on this issue

from different perspectives and designed a series of drought indices to quantify drought. Since different drought indices require different data, handle data in different ways, and have different constraints, the results of different drought indices may vary. Therefore, a single drought index may not be applicable to all drought types and all regions. Theoretically, it is more physically meaningful to use TWS to assess the degree of drought because the soil water content, the total soil deficit, is a more visual representation of its aridity [23] (Zhangli Sun et al., 2018). The method by Thomas et al. [9] (Thomas et al., 2014) defines a drought event as a negative WSD occurring for at least 3 months, but this does not apply to the assessment of all drought events. There are other drought indices that are widely used; for example, the Palmer Drought Index (PDSI) can better monitor regional drought levels and is more responsive to temperature, so it is recognized as a better drought indicator. However, the PDSI still has some limitations, such as the need to determine the water-holding capacity of the soil during the calculation process, the single time scale, etc., and the parameters are difficult to obtain on a large scale. When used for large-scale drought monitoring involving a large area, few stations, and complex surface structure, it will be greatly constrained. In this paper, the variability in water storage is represented by the TWS values recorded by GRACE for each month (e.g., the average of all February months during the study period) [9] (Thomas et al., 2014). However, its spatial resolution (0.25°) is low, and the use of terrestrial water storage deficits alone to determine drought severity may be unreliable because factors other than storage deficits can directly or indirectly interfere with the assessment of drought severity. For example, a large area with slightly lower water storage than the previous average can lead to a large TWS deficit, which may be mistaken for severe drought.

*5.3. WSDI Drought Index Analysis*

The drought intensity can be better shown using the standardized WSDI and drought index. Although the WSD can be used to quantify water storage deficits, it cannot directly assess the strength of drought in different regions [23] (Zhangli Sun et al., 2018). Unlike traditional drought indices, the WSDI is calculated from the change in terrestrial water storage, which can better show the dynamic changes in drought formation. In addition, when there is a need to assess the extent of widespread drought and there is a lack of data from weather or hydrological stations in certain areas, WSDI is based on satellite observations of changes in integrated terrestrial water storage; thus, the amount of water stored can be monitored at any time and is not limited by the availability of traditional observations. Moreover, WSDI data are easy to obtain and simple to calculate compared to traditional drought indices, and the WSDI provides a more intuitive method for drought extent assessment. In conclusion, the WSDI calculated by GRACE provides a valid assessment method for this purpose when assessing the extent of drought over a large area and when relevant site data are missing. Since the index is based entirely on the TWSA observed by GRACE, and the parameters of groundwater storage, evapotranspiration, and soil moisture obtained from GRACE satellite data are in good relative uniformity with the measured data, it can complement ground observations well, but its disadvantage is its low spatial resolution (0.25°). For example, the values of different water reserves are not available in the monthly data of TWSA and are not precise enough when applied to a small area [24] (Emerton et al., 2016). Although the new MASCON solution has a higher spatial resolution than traditional solutions, the WSDI is still more efficient at large spatial scales. Due to its short TWS time series, the monthly average water storage is not precise enough, and it is challenging to assess longer or more precise drought conditions using the WSDI.

## 6. Conclusions

In this study, spatio-temporal and seasonal variations in the arid region in central Asia were analyzed based on TWS changes obtained by GRACE inversion, and the WSDI was calculated from TWS observations. Drought events near the Caspian Sea, Kazakh Hills, Moynkum Desert, and Gurbantünggüt Desert in the arid region of central Asia were

quantified and analyzed over the period of January 2003 to December 2020 (excluding 2017 and 2018 data).

The results show the following: (1) The interannual variation in TWS is obvious in the Caspian Sea region where the TWS value has been decreasing, and the drought degree in the Caspian Sea is continuously increasing. In the northern, southern, and eastern parts of the other three regions, the TWS value has been decreasing in recent years, although the temporal and spatial variation is not apparent. (2) The variation in the TWS in the arid region of central Asia has obvious seasonal characteristics, with the highest TWS value in spring and the lowest TWS value in autumn (autumn is the driest among the four seasons). Besides the Kazakh Hills region in the north of the arid region, the water shortage in the Caspian Sea, Moynkum Desert, and Gurbantünggüt Desert is more severe in autumn and winter than in spring and summer. (3) A total of 22 drought events were observed in the four study areas discussed in this paper. During the major drought, the longest loss period of 95 months (February 2011–December 2020) occurred near the Caspian Sea with a peak loss of −186.43 mm. Through comparison, it was found that some drought events were consistent in time; for example, in the Kazakh Hills, Moynkum Desert, and Gurbantünggüt Desert region in 2008–2010, drought events occurred at the same time, indicating large-scale drought. (4) WSDI trend analysis showed that water stress has worsened in the south and west of the arid region near the Caspian Sea, while water shortage improved in the north.

The method described in this study reliably captured major drought events in large spatial areas. Therefore, it may be an ideal substitute for assessing changes in drought extent over large study areas and when hydrometeorological stations are scarce, providing a scientific basis for adjusting agricultural policies and coping with climate change in arid Asia.

**Author Contributions:** J.L. and G.Z. conceived the idea of the study; K.Z. and Y.J. conducted formula analysis and methodological investigation; Y.L. and M.Y. conducted data curation; J.L. wrote the paper and organized software debugging; W.Z. and D.Q. carried out relevant calculation validation; X.L. and L.Y. checked and edited language. All authors have read and agreed to the published version of the manuscript.

**Funding:** This research was financially supported by the National Natural Science Foundation of China (41867030, 41971036) and the National Natural Science Foundation innovation research group science foundation of China (41421061). The authors are very thankful for the support of the above funds.

**Data Availability Statement:** The data that support the findings of this study are available on request from the corresponding author. GRACE data can be obtained from the space research center at the International Center for Global Gravity Field Models (ICGEM) (http://icgem.gfz-potsdam.de/series (accessed on 20 May 2022).

**Acknowledgments:** Thanks to the International Center for Global Gravity Field Models (ICGEM) for the GRACE data.

**Conflicts of Interest:** The authors declare no conflict of interest.

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
