# Peer review of "GRACE Combined with WSD to Assess the Change in Drought Severity in Arid Asia"

_remotesensing, doi:10.3390/rs14143454_

Round 1

Reviewer 1 Report

Abstract should be compiled more concise. 1. What have you done? 2. How have you done? 3. Where have you done it? 4. What is the main conclusion? 5. Why is it important in wider aspect? – the present abstract, the above points 4 and 5 are missing (the first sentence – which I suggest to delete – is far from Point5, and the Conclusions of the paper (Page 11) are different from the figures in the abstract.

Abbreviations – hard to follow them for wider remote sensing audience. Defining abbreviation solutions just in the abstract is less favorable, do it in first occurrence in main text, and again, at key points (e.g. equations).

My main problems are around the Equation (1). Not with the content – it is and old and well-known formula – however:

-      -are the Δη values the same as TWSA values in Equation (2)? If yes, please use this. If not, please specify the relation.

-      -Eq (1) says the left side is a function of time. Where occurs this in the right side? (Of course, Clm and Slm are the functions of time, please indicate it.

-      -θ is the polar distance, not the latitude.

-      -Please indicate the longitude its native sign, λ (instead of the native sign of the latitude, the φ)

-      -l and m are respectively the order and DEGREE of (not the “number”) of the SH.

-     -What is k1? (not defined)

-     -Plm are the fully normalized Legendre functions, independently of the argument (cos θ).

To Point 2.1 (Research data): why is the expression/abbreviation “mascon” in the downloadable data. Is it standing for “mass concentrations”. (it would be quite obvious.) If you, please indicate it, and also give a short explanation, why is it applied in water balance studies.

To Discussion. Hm, I assume that more draughts cause severe problems to 50M farmers, however a scientific discussion could (or here: would) be of two kinds:

1.       How the present data are supporting of denying our previous knowledge (a) of the method (b) of the draught situation of the area?

2.       Can the remotely sensed data (GRACE-derived variations of potential surface changes) connected to independent, locally observed data, such as precipitation or harvest yields?

Without this, the Conclusions are – albeit of being real ones – quite thin to share to the scientific community.

Author Response

Thank you for your letter and for the reviewer’s comments concerning our manuscript entitled “GRACE combined with WSD to assess the change of drought severity in arid Asia” (Manuscript ID: remotesensing-1765618).

According to the comments of the reviewer, we have revised our manuscript carefully. The revised portions have been marked in red in the manuscript changes version. The main corrections and the response to the reviewers’ comments are as follows.

Reviewer: 

Point 1: your manuscript should undergo extensive English revisions.

Response 1: Thank you very much for your suggestion. We have embellished the language.

Point 2: Abstract should be compiled more concise. 1. What have you done? 2. How have you done? 3. Where have you done it? 4. What is the main conclusion? 5. Why is it important in wider aspect? – the present abstract, the above points 4 and 5 are missing (the first sentence – which I suggest to delete – is far from Point5, and the Conclusions of the paper (Page 11) are different from the figures in the abstract. 

Response 2: Thank you very much for your suggestion. We have revised the Abstract,In particular, the conclusions and implications of the paper are clarified.

  1. What have you done?

In this study, we quantified drought severity based on land terrestrial water storage (TWS) changes in GRACE data.

  1. How have you done?

We used the Water Storage Deficit (WSD) and Water Storage Deficit Index (WSDI) to identify the drought events and evaluate the drought severity. The WSDI calculated by GRACE provides an effective assessment method when assessing the extent of drought over large areas under a lack of site data.

  1. Where have you done it?

Water Storage Deficit Index (WSDI) to identify the drought events and evaluate the drought severity in arid Asia.

  1. What is the main conclusion?

The results show a total of 22 drought events in the Central Asian dry zone during the study period. During spring and autumn, the droughts among these incidents occurred more frequently and severely. The longest and most severe drought occurred near the Caspian Sea. In the arid area of central Asia, the north of the region tended to be moist (the WSDI value was 0.04 yr−1), and the south, east, and Caspian Sea area tended to be drier (the WSDI values were -0.07yr−1 in the south, -0.11 yr−1 in the east, and -0.19 yr−1 in the Caspian Sea).  

  1. Why is it important in wider aspect?

These study results can provide a key scientific basis for agricultural development, food security, and climate change response in the Asian arid zone.

Point 3: Abbreviations – hard to follow them for wider remote sensing audience. Defining abbreviation solutions just in the abstract is less favorable, do it in first occurrence in main text, and again, at key points (e.g. equations).

Response 3: Thank you very much for your suggestion. We have revised this. As shown below:

On page 1: The terrestrial water storage (TWS) derived from GRACE is an important method for quantifying changes in surface water and groundwater storage on land. 

On page 3:Therefore, terrestrial water storage anomalies (TWSA) on land can be directly described by gravity coefficient anomalies (,).

On page 3: The Water Storage Deficit (WSD) can be calculated by obtaining the TWSA time series from GRACE [9](Thomas et al., 2014)

On page 3: To better assess WSD-based drought, this parameter translates to the Water Storage Deficit Index (WSDI) as follows.

Point 4: My main problems are around the Equation (1). Not with the content – it is and old and well-known formula – however:

-      -are the Δη values the same as TWSA values in Equation (2)? If yes, please use this. If not, please specify the relation.

Response 4: First, equation (1) is my attempt to briefly explain the principle of GRACE gravity satellite inversion of terrestrial water storage. Secondly, On page 3 I add a description of Δη, which represents the Equivalent Water Height (EWH). TWSA can be directly described by Δη, but here I just want to briefly describe the principle of GRACE gravity satellite inversion of terrestrial water storage, so to have some distinction, Δη is used for the description. We have added on page 3: Δη is the Equivalent Water Height (EWH).

Point 5: Eq (1) says the left side is a function of time. Where occurs this in the right side? (Of course, Clm and Slm are the functions of time, please indicate it.

Response 5:We are very sorry, I made a writing error here, after I researched the literature again, there is no ‘t’ on the left side of the equation. we have modified the equation. As shown below:

     (1)

Point 6: θ is the polar distance, not the latitude.

Response 6: We are very sorry, we have corrected the error. As shown below:

θ  is the polar distance.

Point 7: Please indicate the longitude its native sign, λ (instead of the native sign of the latitude, the φ).

Response 7: We have made changes to the errors. As shown below:

      (1)

λ is longitude

Point 8: l and m are respectively the order and DEGREE of (not the “number”) of the SH.

Response 8: We have corrected the error. As shown below:

l, m are respectively the order and DEGREE of the spherical harmonic function.

Point 9: What is k1? (not defined).

Response 9: k1 is the first-order load Love number, and k1 has been described in the supplement on page 3.

Point 10: Plm are the fully normalized Legendre functions, independently of the argument (cos θ).

.Response 10: We have corrected the error. As shown below:

Plm  is a fully normalized Legendre function.

Point 11: To Point 2.1 (Research data): why is the expression/abbreviation “mascon” in the downloadable data. Is it standing for “mass concentrations”. (it would be quite obvious.) If you, please indicate it, and also give a short explanation, why is it applied in water balance studies.

.Response 11: First of all, the download data was pasted wrongly due to my carelessness. International Center for Global Gravity Field Models (ICGEM) (http://icgem.gfz-potsdam.de/series), the website where the data is downloaded, requires that the data to be used refer to the references specified by it, which We have modified. As shown below:

The space research center at the University of Texas (CSR), the German Research Centre for Geosciences (GFZ), and the Jet Propulsion Laboratory (JPL) provide three GRACE/GRACE-FO spherical harmonic (SH) solution land storage (RL06) datasets that were used in this study [12] (J.Kusche et al.,2009).

Secondly, mascon is actually a method, the mass concentration method, which makes data that is a product that can be used directly.

Point 12: To Discussion. Hm, I assume that more draughts cause severe problems to 50M farmers, however a scientific discussion could (or here: would) be of two kinds:

How the present data are supporting of denying our previous knowledge (a) of the method (b) of the draught situation of the area?

Response 12: After our references research, we found that the figure of "50M" here is inaccurate and, in fact, most developing countries in the region lack accurate demographic information,and we have revised it and added references to it later. As shown below:

directly threatening the livelihoods of hundreds of millions of local farmers and herdsmen [19](Troy S., 2018).

Point 13: 2. Can the remotely sensed data (GRACE-derived variations of potential surface changes) connected to independent, locally observed data, such as precipitation or harvest yields?

Without this, the Conclusions are – albeit of being real ones – quite thin to share to the scientific community.

Response 13: 

  • Comparison with the measured precipitation data from China Meteorological Data Network, the trend is consistent.;As shown in the figure below:

  • Comparative analysis with different normalized drought indices showed consistent trends;

The WSDI drought index has been compared with different drought indices in many references, such as SPEI, PDSI, SRI, etc. The results can confirm the reliability of the WSDI drought index. The following are some relevant references:

[1] Sinha D ,  Syed T H ,  Famiglietti J S , et al. Characterizing Drought in India Using GRACE Observations of Terrestrial Water Storage Deficit[J]. Journal of Hydrometeorology, 2017, 18(2):381-396.

[2]C Uzcátegui-Briceo. Drought Assessment in the So Francisco River Basin Using Satellite-Based and Ground-Based Indices[J]. Remote Sensing, 2021, 13.

We believe that the physical mechanism of the GRACE combined with WSD method is clearer and will become a more scientifically accurate method to assess drought with the advancement of remote sensing technology.

Reviewer 2 Report

The manuscript GRACE combined with WSD to assess the change of drought 

severity in arid Asia written by Jiawei Liu , Guofeng Zhu , Kailiang Zhao , Yinying Jiao , Yuwei Liu , Mingyue Yang  ,Wenhao Zhang , 

Dongdong Qiu , Xinrui Lin    and Linlin Ye  is about  a quantification of drought severity based on changes in land terrestrial water storage (TWS).

Please correct these:

Please change overview of the study area with Study area

Please change Research methods and data sources with Material and methods and move it before study area.

paragraph 2.1, please move web links in the references

Please add a bullet for formula 1 symbols

Please change 3. Results and analysis in Results

Please modify the equations in trend line to a more human readable format: i.e from of 0.1897x+381.6433 to 0.19x+381.64

Author Response

Thank you for your letter and for the reviewer’s comments concerning our manuscript entitled “GRACE combined with WSD to assess the change of drought severity in arid Asia” (Manuscript ID: remotesensing-1765618).

According to the comments of the reviewer, we have revised our manuscript carefully. The revised portions have been marked in red in the manuscript changes version. The main corrections and the response to the reviewers’ comments are as follows.

Reviewer: 
Point 1: Please change overview of the study area with Study area. 

Response 1: We have changed the overview of the study area to the Study area. As shown below:

2.Study area

Point 2: Please change Research methods and data sources with Material and methods and move it before study area.

Response 2: Thank you very much for your suggestion. We have revised this. As shown below:

1.Material and methods. 

  1. Material and methods

1.1. Research Data

The space research center at the University of Texas (CSR), the German Research Centre for Geosciences (GFZ), and the Jet Propulsion Laboratory (JPL) provide three GRACE/GRACE-FO spherical harmonic (SH) solution land storage (RL06) datasets that were used in this study [12] (J.Kusche et al.,2009). The water storage data obtained from these datasets are expressed in centimeter Equivalent Water Height (EWH) [13] (Landerer and Swenson, 2012). The three organizations performed several post-processing corrections to their SH solutions to isolate surface impound signals. The C20 spherical harmonic coefficient was replaced by satellite laser-ranging solutions [14] (Cheng, 2011). A method by Swenson et al. [15](Swenson et al., 2008) was used to estimate the first-order coefficients. A 300 km wide Gaussian filter was used to smooth the spherical harmonic solution to minimize correlation and high-frequency error. To reduce the error and make the data more accurate, the monthly TWS data provided by CSR, GFZ, and JPL were processed, extracted, and added to obtain the average value. All the missing data were filled with the average values of the months before and after the missing month using the linear interpolation method. GRACE returned to Earth in July 2017, and the second-generation GRACE Follow (GRACE-FO) started operations in May 2018. To assess the drought situation more accurately in the arid regions of central Asia, in this paper, we used data for 16 years from January 2003 to December 2020, excluding 2017 and 2018. We defer to the meteorological division law, namely, that the spring months are March, April, and May; the summer months are June, July, and August; the autumn months are September, October, and November; and the winter months are December, January, and February.

1.2. GRACE inversion

The principle of GRACE satellite data inversion is to calculate the change in the Earth's mass density at each moment by calculating the change in the spherical harmonic potential coefficient, which describes the time-varying gravity field and expresses the change in the mass density in the form of the equivalent water height. The Earth's gravity field is often expressed in terms of the spherical harmonics (SH) expression of the geoid that is the equipotential surface corresponding to the mean sea level. In general, the spherical harmonic expression of the geoid serves as the Earth's gravity field by the equipotential surface. Most remaining signals are relevant to changes in the TWS after removing atmospheric and ocean mass effects from the climate and ocean general circulation models. Therefore, terrestrial water storage anomalies (TWSA) on land can be directly described by gravity coefficient anomalies (,):

     (1)

where is the Equivalent Water Height (EWH); is the average radius of the Earth;  and  are the variation values of spherical harmonic potential coefficients of the Earth's gravitational field;  is water density;  represents the average density of the Earth; and  are, respectively, the order and DEGREE of the spherical harmonic function; , are the regularized spherical harmonic potential coefficients in the Earth's gravitational field;   is the polar distance;  is longitude;  is a fully normalized Legendre function; and  is the first-order load Love number.

1.3. Estimation of insufficient water storage based on GRACE

The Water Storage Deficit (WSD) can be calculated by obtaining the TWSA time series from GRACE [9](Thomas et al., 2014), as shown below:

      (2)

In the formula,  denotes month j of year i of the TWSA time series obtained by GRACE, and is the long-term (January 2003 to December 2020) mean of TWSA for the same month (the average of 16 values in month j). A negative WSD means that, compared with the monthly average, insufficient land water storage has led to a water storage deficit, while a positive value means that there is surplus water storage. A drought event is defined as a negative WSD lasting three months. To better assess WSD-based drought, this parameter translates to the Water Storage Deficit Index (WSDI) as follows: 

                             (3)

where is the mean value of the WSD time series, and  represents the standard deviation of the WSD time series. The magnitude of the WSDI represents the drought intensity, and the classification of the drought intensity is shown in Table 1.

Table 1. Characterization of drought intensity by WSDI.

WSDI value

Drought category

W>0

No drought

0≥W>-1

Mild drought

-1≥W>-2

Moderate drought

-2≥W>-3

Severe drought

-3≥W

Extreme drought

  1. Study area

The arid region of central Asia (34°34 '-55° 43'N, 46°48' -106° 98'E) is located north of Pamir/the Qinghai–Tibet Plateau, south of the Urals/Altai Mountains, and east of the Caspian Sea and Volga River, and extends to Helan Mountain/Wushaoling mountains. It is the widest arid region in the temperate zone of the Earth's northern hemisphere [16] (Chen Xi et al., 2013). The arid areas are located in the hinterland of Eurasia, with a low proportion of maritime airflow reaching them. The upper westerlies transport weak water vapor from the remote Arctic and Atlantic Oceans. Due to the interception and uplift of vertical terrain, precipitation is concentrated in the mountains, and snow glaciers form in the mountains. The average annual precipitation in arid areas is less than 150 mm, and the spatial distribution is extremely unbalanced. The annual precipitation in mountainous areas is more than that in basins and plains [17] (Balashova Y et al.,2006). The annual precipitation in forest areas of the Tianshan Mountains and the Altai Mountains reaches 1000 mm [18](Hu Ruji et al.,2004). The annual precipitation in Turpan and Hami Basin is less than 100 mm [17](Balashova Y et al.,2006).

Fig.1 Overview map of the study area in the arid region of central Asia

Point 3: paragraph 2.1, please move web links in the references

Response 3: First of all, the download data was pasted wrongly due to my carelessness. International Center for Global Gravity Field Models (ICGEM) (http://icgem.gfz-potsdam.de/series), the website where the data is downloaded, requires that the data to be used refer to the references specified by it, which We have modified. As shown below:

The space research center at the University of Texas (CSR), the German Research Centre for Geosciences (GFZ), and the Jet Propulsion Laboratory (JPL) provide three GRACE/GRACE-FO spherical harmonic (SH) solution land storage (RL06) datasets that were used in this study [12] (J.Kusche et al.,2009).

Point 4: Please add a bullet for formula 1 symbols

Response 4: We have revised formula 1 symbols. As shown below:

     (1)

where is the Equivalent Water Height (EWH); is the average radius of the Earth;  and  are the variation values of spherical harmonic potential coefficients of the Earth's gravitational field;  is water density;  represents the average density of the Earth; and  are, respectively, the order and DEGREE of the spherical harmonic function; , are the regularized spherical harmonic potential coefficients in the Earth's gravitational field;   is the polar distance;  is longitude;  is a fully normalized Legendre function; and  is the first-order load Love number.

Point 5: Please change 3. Results and analysis in Results

Response 5: Thank you very much for your suggestion. We have made the changes as you requested. As shown below:

  1. Results

Point 6: Please modify the equations in trend line to a more human readable format: i.e from of 0.1897x+381.6433 to 0.19x+381.64

Response 6: We have modified the equations in the trendlines to a more readable format. As shown below:

On page 1: In the arid area of central Asia, the north of the region tended to be moist (the WSDI value was 0.04 yr−1), and the south, east, and Caspian Sea area tended to be drier (the WSDI values were -0.07yr−1 in the south, -0.11 yr−1 in the east, and -0.19 yr−1 in the Caspian Sea).

On page 8: The WSDI in the northern part of the arid area of central Asia shows an overall upward trend (0.04 yr)1), while those for the Caspian Sea (-0.19 yr1), south (-0.07 yr1), and eastern (-0.11 yr1) areas show a decreasing tendency.

Reviewer 3 Report

Dear authors, 

Some small comments on your paper:

1.      GRACE signal was transformed into variation (changes) of the Total Water Storage Anomaly (TWSA) – page 1. That means that the real Terrestrial Water Storage (TWS) is not known. On the other hand, at page 10 it is written that the parameters of water storage obtained from GRACE satellite data are in good agreement with the measured data. Groundwater storage, soil moisture parameters etc are difficult (if not impossible) to be evaluated at a regional scale by direct measurements. Please, clarify.

2.      In the formula (2) the WSD is obtained at the level of each month. However, in Figure 3, the results are presented at the seasonal level. Probably, average values were obtained for the 3 months of every season. Please, clarify.

3.      Paragraph 3.1 – page 5: It would be good to present the spatial resolution (especially because at page 10 it is written that the spatial resolution is low).

4.      If WSD calculation is based on low spatial resolution, how WSDI (which is derived from WSD) can provide better results and can better show the dynamic changes (page 10) of drought formation? Please, clarify.

5.      Figure 4 – Both blue and orange bars represent WSD (not only the blue ones). The blue bars correspond to water excess above the average value of the corresponding month. May be the legend for the blue bars could be: “WSD above the long-term mean”, or “Above norm” or “No drought”.

6.      Large parts of Chapter 4 can be moved to Introduction.

Best regards

Author Response

Thank you for your letter and for the reviewer’s comments concerning our manuscript entitled “GRACE combined with WSD to assess the change of drought severity in arid Asia” (Manuscript ID: remotesensing-1765618).

According to the comments of the reviewer, we have revised our manuscript carefully. The revised portions have been marked in red in the manuscript changes version. The main corrections and the response to the reviewers’ comments are as follows.

Reviewer: 
Point 1: GRACE signal was transformed into variation (changes) of the Total Water Storage Anomaly (TWSA) – page 1. That means that the real Terrestrial Water Storage (TWS) is not known. On the other hand, at page 10 it is written that the parameters of water storage obtained from GRACE satellite data are in good agreement with the measured data. Groundwater storage, soil moisture parameters etc are difficult (if not impossible) to be evaluated at a regional scale by direct measurements. Please, clarify. 

Response 1: I'm very sorry, it's due to my language that I didn't express myself clearly. Firstly, GRACE extracts the variability of the Total Water Storage Anomaly (TWSA), which is used to calculate the Water Storage Deficit Index (WSDI). Secondly, At page 9 it is written that he parameters of groundwater storage, evapotranspiration and soil moisture obtained from GRACE satellite data are in good relative uniformity with the measured data, it means the changes are relatively consistent. We have modified this. As shown below:

soil moisture obtained from GRACE satellite data are in good relative uniformity with the measured data.

Point 2:  In the formula (2) the WSD is obtained at the level of each month. However, in Figure 3, the results are presented at the seasonal level. Probably, average values were obtained for the 3 months of every season. Please, clarify.

Response 2: It is sure that the WSD is obtained at the level of each month, the seasonal data is obtained by adding up the 3 months and taking the average, for example, spring for March, April, May data is added up and divided by 3.:

Point 3: Paragraph 3.1 – page 5: It would be good to present the spatial resolution (especially because at page 10 it is written that the spatial resolution is low).

Response 3: Thank you very much for your suggestion. We have revised this. As shown below:

On page 10: However, its spatial resolution (0.25°) is low

Point 4: If WSD calculation is based on low spatial resolution, how WSDI (which is derived from WSD) can provide better results and can better show the dynamic changes (page 10) of drought formation? Please, clarify.

Response 4: First, it is because of its low resolution that WSDI is mentioned in Discussion 3.3 as being more suitable for large scale drought assessment. Secondly, GRACE detects vertically integrated water storage changes from the land surface to the deepest aquifers, thus providing more information that can reveal the dynamic changes of drought formation.

Point 5:  Figure 4 – Both blue and orange bars represent WSD (not only the blue ones). The blue bars correspond to water excess above the average value of the corresponding month. May be the legend for the blue bars could be: “WSD above the long-term mean”, or “Above norm” or “No drought”.

Response 5: Thank you very much for your suggestion. We have revised the WSD to 'No drought'.. As shown below:

Point 6:  Large parts of Chapter 4 can be moved to Introduction..

Response 6: Thank you very much for your suggestion. We have added some chapter 4 content in the introduction section. As shown below:

Drought is one of the most important natural disasters in the world; it not only affects human activities but also has a negative impact on the environment, agriculture, and economic development. Monitoring and quantifying drought-induced changes in water storage are of positive significance for assessing the extent of drought. Moreover, under the current rapid economic development and population increase scenario, a more scientific assessment of the change in drought extent is necessary [1](Long et  al., 2013). Water shortages will be even worse in a decade’s time, affecting as much as half the world's population [2] (Dharpure et al., 2020). Currently, climate change is becoming more unstable, and the negative effects of drought are likely to intensify [3](Gerdener et al., 2020). The Asian arid zone is located in the hinterland of the continent, with less drought and more rain, including the widespread Gobi desert. The ecological environment here is very fragile, and climate change is very significant. The Asian arid zone has been under serious pressure from climate change and the frequent occurrence of extreme climate events [4] (Trenberth et al., 2014). Therefore, it is very important to accurately assess the degree of aridity in arid and semi-arid regions.

However, the quantification of drought and its hydrological effects remains a major challenge due to the limitations of monitoring means. The Gravity Recovery and Climate Experiment (GRACE) launch generated a new tool for monitoring and assessing drought events. After GRACE observations, it was concluded that the gravity change mainly resulted from the large-scale movement of surface water, and the GRACE signal was transformed into variation in the Total Water Storage Anomaly (TWSA) on land [5](Hu et al., 2019). The terrestrial water storage (TWS) derived from GRACE is an important indicator for quantifying changes in surface water and groundwater storage on land [6] (Wahr and John, 2004) and is widely used to monitor hydrological changes caused by drought. The TWSA retrieved by GRACE includes all forms of water stored above and below the surface, including snow, surface water, soil moisture, and groundwater [7] (Wu et al., 2021), and can thus serve as an effective substitute for hydrological information.

Based on the GRACE TWSA, several drought indices have been put into use and developed to date to elaborate drought assessment. Yirdaw et al. [8] (Yirdaw et al., 2008) proposed the total storage deficit index (TSDI) by using the TWSA value of GRACE and the Palmer drought Severity Index (PDSI) and soil water deficit index (SMDI). Thomas et al. [9](Thomas et al., 2014) proposed a quantitative method to measure the occurrence and severity of hydrological drought based on GRACE data. Sinha et al. [10](Sinha et al., 2017) further extended the method of measuring the occurrence of hydrological drought based on GRACE data and designed a drought index, the Water Storage Deficit Index (WSDI), using the TWS changes estimated by GRACE to quantify the intensity and severity of drought. Hosseini-moghari et al. [11] (Hosseini-moghari et al., 2019) developed an improved total storage deficit index (MTSDI) using the remaining time series of TWSA to eliminate the influence of human factors on TWSA changes. Unlike the traditional drought index, the WSDI is calculated from the change in terrestrial water storage, which can better show the dynamics of drought formation. In addition, when there is a need to assess the extent of drought over a large area and there is a lack of data from meteorological or hydrological stations in some areas, the WSDI is based on the integrated land water storage changes observed by satellites, so the storage amount can be monitored at any time without the limitations of traditional observations.

In recent years, drought has become more severe and has frequently occurred in many parts. In this study, we evaluated the degree of drought in Asia and Europe based on GRACE data. Our aim was to (1) clarify the overall change trend of the drought degree in different regions of arid Asia and to (2) analyze the influencing factors leading to the change in drought degree. As a classical case, this study provides an evaluation of the drought degree in an arid region, which can form a scientific basis to facilitate adjusting agricultural policy and coping with climate change in that area.
